# Is Parental Mathematics Anxiety Associated with Young Children's Arithmetical Performance?

Elien Bellon [1,*], Elsje van Bergen [2,3] and Ann Derore Dowker [4]

[1] Department of Parenting and Special Education, University of Leuven, 3000 Leuven, Belgium
[2] Department of Biological Psychology, Vrije Universiteit Amsterdam, 1081 BT Amsterdam, The Netherlands
[3] LEARN! Research Institute, 1081 BT Amsterdam, The Netherlands
[4] Department of Experimental Psychology, University of Oxford, Oxford OX2 6GG, UK
[*] Correspondence: elien.bellon@kuleuven.be

**Abstract:** It has been suggested that parental mathematics anxiety may influence their children's mathematics anxiety, attitudes, and performance. It remains an open question whether these parent-child associations differ by parental sex or parental involvement. We tested 249 Dutch-speaking Belgian participants, forming 83 (biological) mother–father–child trios. The 83 children (age: $M = 5.74$; $SD = 0.30$) attended Kindergarten. We tested their nonsymbolic comparison, symbolic numerical magnitude processing, numeral recognition, arithmetic, and matrix reasoning. We assessed both parents' arithmetic skills, math anxiety, educational level, and division of care. More math-anxious parents tended to be less highly educated ($r\sim0.42$) and poorer at math ($r\sim0.30$). Compared to fathers, mothers had lower arithmetic, higher math anxiety, and higher educational level. Assortative mating (i.e., a significant spousal correlation) was found for educational level and arithmetic. Mothers' (but not fathers') educational level predicted children's arithmetic scores ($r = 0.31$). Other parent-offspring correlations were non-significant. Most of the children's test scores are intercorrelated. The parental characteristic that best predicted five- and six-year-olds' arithmetic performance was maternal educational level rather than mathematical anxiety or performance. We discuss these findings in relation to the used measures, parental gender and involvement, children's age, statistical power, and genetic and environmental transmission. The field is just starting to understand whether and how mathematics anxiety and the skills of parents influence those of their offspring.

**Keywords:** arithmetic; mathematics; math anxiety; intergenerational transmission; assortative mating; kindergarten; parent–child associations

## 1. Introduction

Many children and adults have highly negative attitudes to mathematics, including significant levels of anxiety about the subject [1–9]. Most studies suggest that mathematics anxiety increases with age [10–14] though there is increasing evidence that it can be found even in the early primary school years [15–20].

Some studies have shown that parents and the home numeracy activities that they provide correlate with children's mathematical development. Some suggested that parental mathematics anxiety may influence their children's attitudes to and performance in mathematics [21–23]. On the whole, parental anxiety and attitudes seem to be more associated with children's mathematics anxiety than with the children's actual performance [23,24], though they may be associated with both.

Parental mathematics anxiety could affect children's attitudes and performance in several ways. It may lead them to avoid mathematics as much as possible. While it is generally assumed that mathematics anxiety will lead to mathematics avoidance, this has not usually been tested. However, a recent study by Choe, Jenifer, Rozek et al. [25] did indeed show that mathematics anxiety was associated with a tendency to select easier, low-reward mathematics problems over more difficult, high-reward mathematics problems.

Parental mathematics avoidance may limit their provision of home numeracy activities for their children. More broadly, it may have limited their own educational and employment opportunities, often resulting in lower socio-economic status for the parents and, thus, for their children.

Parental mathematics anxiety may also result in the transmission to their children of negative attitudes to mathematics and in the children's coming to associate mathematics with negative emotions. The influence of parents' mathematics anxiety on their children seems to depend, at least in part, on the extent to which the parents are involved with their children's mathematical activities. Maloney et al. [22] found that parents' mathematics anxiety was negatively associated with their children's mathematics progress in first and second grade if the parents frequently helped their children with mathematics homework but had little association with their children's progress if they did not frequently assist with mathematics homework. The effect of parental mathematics anxiety appeared to be specific to mathematics; it did not predict children's reading achievement. The results may be due to parents transmitting their attitudes to their children; or due to other environmental or genetic factors influencing both parents' and children's attitudes and performance in mathematics. Alternatively, it may be because parental mathematics anxiety is associated with parents anticipating that their children are likely to fail in mathematical tasks, putting pressure on the children and expressing anger, frustration, and other negative emotions toward them in mathematics-related situations. Di Stefano et al. [26] obtained some evidence for the latter explanation: they found that mathematics anxiety in parents was associated with their self-reported level of conflict, stress, frustration, and emotionality when helping children with their mathematics homework. Such parental frustration and emotionality could well cause or contribute to children's developing negative reactions to mathematics. Both the amount of parental input and the type of task involved (e.g., formal versus informal; associated versus not associated with the external assessment) may affect the level of parental emotionality to which children are exposed, and these, in turn, are likely to vary with the child's age.

In addition, the effects of parental anxiety on children's performance may vary with culture and with socio-economic status. Indeed, there is evidence for ethnic group differences in the effects of different types of parental input on children's mathematics achievement within an American context [27]. There is much converging evidence that there are significant cultural and national differences in mathematics anxiety (e.g., [28,29]), although there appears to be a negative relationship between mathematics anxiety and mathematics performance in virtually all places and cultures [13,29].

There are also significant cultural differences in broader attitudes to mathematics. For example, East Asians seem to place a higher value on the importance of mathematics and seem to be more likely to attribute success and failure in mathematics to effort as opposed to ability than Europeans or Americans [30,31], with the caveat that differences can be very marked between groups *within* a culture or geographical area [30]. It might therefore be that the effects of parental mathematics anxiety on the relationships between parental input and children's attitudes and performance might vary between cultures: for example, mathematics anxiety in East Asian parents might be more likely to lead to excessive parental pressure on children, and mathematics anxiety in Western parents might be more likely to lead to parents transmitting dislike and avoidance of mathematics to their children.

Socio-economic status is also likely to have a significant influence on parental attitudes and input and their effects on children's mathematics achievement. In particular, it is found very consistently, across many countries, that parents of lower SES have significantly higher levels of mathematics anxiety than parents of higher SES [32], likely both as cause and effect.

Casad, Hale, and Wachs [21] found, in a study of teenagers and their parents, that there was a significant association between parents' and children's mathematics anxiety. This association was particularly strong between parents and children of the same gender, especially between mothers and daughters. However, this study, like most studies of associations between parents' and children's mathematics anxiety, included only one

parent, usually the mother, and did not compare mothers and fathers. There is evidence that mothers' and fathers' attitudes and anxiety can have different associations, partly as a function of the age and gender of the children. Szczygiel [24] found that mathematics anxiety in fathers was associated with mathematics anxiety in both girls and boys in first grade and in girls but not boys in third grade. Mathematics anxiety in mothers was associated with third-grade children's mathematics anxiety but not that of younger children.

Del Rio, Susperreguy, Strasser, and Salinas [33] carried out a study in Chile. They did not find a significant direct association between parents' levels of mathematics anxiety and their children's mathematics performance. However, parental mathematics anxiety was negatively associated with the extent of parental formal home numeracy practices, and children's mathematical performance was significantly associated with their mothers' (but not fathers') formal numeracy practices.

There are several reasons why there might be differences in associations between children's mathematical performance and their mothers' versus fathers' mathematics anxiety. One is that mothers commonly spend more time with their children than fathers do [34]. Another is that, on the whole, mothers are likely to be more anxious about mathematics than fathers are, as mathematics anxiety tends to be higher in females than males [4,35]. Another reason, possibly related to the above, is that mathematics anxiety may interact with gender stereotypes about mathematics [36–38].

Dowker [39] has suggested that it is important to investigate whether parental mathematics anxiety is associated with children's mathematical development as early as the preschool years, either by affecting parents' provision of home-numeracy activities or by the intergenerational transmission of negative emotions. There have been very few studies with such young children, and they have given conflicting results. Becker, Litkowski, Duncan et al. [40] carried out one of the very few studies on this topic and found that parents' mathematics anxiety was indeed significantly negatively related to both their sons' and daughters' mathematics performance during the spring of their pre-kindergarten year, even after controlling for the children's mathematics performance at the beginning of the year. By contrast, Cosso et al. [41] did not find a relationship between preschoolers' numerical abilities and their parents' mathematics anxiety.

One important issue that has rarely been examined is whether fathers' and mothers' mathematics anxiety are correlated [23]. Such a correlation between spouses is called assortative mating. Assortative mating could be because spouses actively match on a trait (termed phenotypic assortment), spouses meet each other in a certain community, for example, at university (termed social homogamy), or because partners become more similar over time, for example, due to mutual communication and transmission of attitudes and emotions toward mathematics (termed spousal convergence). It is also important to examine correlations between parental mathematics anxiety and arithmetical performance and educational level. Such correlations might be expected as many studies indicate a negative relationship between mathematics anxiety and mathematical performance [8,13,29,42], and it is also likely that the avoidance of mathematics may restrict educational opportunities more generally. It is, therefore, possible that some of the findings about relationships between parental mathematics anxiety and children's mathematical performance are, in fact, due to associations with parental educational level and/or arithmetical performance. They were therefore assessed in the present study.

The main topics for investigation in the present study are:

(1) Differences between fathers and mothers in arithmetical performance, mathematics anxiety, and educational level. Given earlier findings regarding gender differences, it was expected that mothers would show greater mathematics anxiety but that there would be few differences in arithmetical performance or educational level.

(2) Correlations within parents between arithmetical performance, mathematics anxiety, and educational level. It was expected that, in both mothers and fathers, arithmetical performance would correlate positively with educational level and that both arithmetical performance and educational level would correlate negatively with mathematics anxiety,

(3)     Correlations between fathers and mothers (i.e., assortative mating) with regard to arithmetical performance, mathematical anxiety, and educational level. It was expected that there would be positive correlations between fathers and mothers for all these factors, as had already been found with regard to parents of sixth-grade pupils [23].

(4)     Correlations between different performance measures in children. It was expected that arithmetic would correlate with numeral recognition, symbolic number comparison speed, and nonsymbolic number comparison speed, which would all correlate with one another, and that it would show a stronger relationship to symbolic than nonsymbolic number comparison speed. It was also expected to correlate with a measure of nonverbal intelligence: the Matrix Reasoning test.

(5)     Associations between parental characteristics and children's arithmetic and other performance measures. It was expected that parents' educational level would be positively associated with all mathematical performance measures in their children and with the children's matrix reasoning and that there would be more specific associations positively between children's arithmetic and their parents' arithmetic score and negatively between children's arithmetic and their parents' mathematics anxiety score. It was expected that while children's numerical measures, in general, would correlate with their parents' mathematics anxiety, the correlations would be higher for measures recognizably involving arithmetic itself rather than for symbolic magnitude processing, numeral recognition, or especially nonsymbolic magnitude comparison. It was also expected that children's performance would be more associated with their mothers' than their fathers' characteristics, but that this would be influenced by the relative amount of time that children spent with their mothers and their fathers. It was important to take into account the possibility that any differences found between fathers' and mothers' level of association with children's performance might be due not to gender effects as such but to the fact that one parent, most commonly the mother, often spends more time with the children [34]. Therefore, in the present study, as in the earlier study of sixth-grade pupils [23], we sought to investigate which of the parents had a greater caring role and whether this moderates the level of association between parental characteristics and children's performance.

## 2. Method

### 2.1. Participants

The present study included 249 participants. These were 83 kindergarten children (45 girls, 38 boys; mean age 5.73 years; s.d. 0.30); their 83 mothers (mean age 35.69 years; s.d. 4.19) and their 83 fathers (mean age 38.51 years; s.d. 5.20. Parents of all children received an information sheet on the study and provided written informed consent for the participation of the family (child and both parents). In all 83 families, the child was living with both biological parents. In addition, all 83 families met the following by applying the following criteria, similar to those set by Vanbinst et al. [23]: (1) Both parents gave informed consent for their own and their child's participation; (2) All tasks and questionnaires were completed by the child and both parents; (3) None of the children was repeating kindergarten; (4) All children were taught at school in Dutch from the age of 2 years and 6 months, and both parents mastered the Dutch language sufficiently to complete the arithmetic test and mathematics anxiety questionnaire without difficulties. The study and consent procedures were approved by the Social and Societal Ethics Committee of the University of Leuven, Belgium (G-2016 03 533).

### 2.2. Materials, Tasks and Questionnaires

2.2.1. Children

Nonsymbolic Comparison

Nonsymbolic numerical magnitude processing skills were measured with a comparison task in which kindergartners had to indicate the larger of two presented dot arrays [43]. The number of dots per array varied from 1 to 9. A trial started with a 200 ms fixation

point in the center of the screen. The dot arrays were generated with the MATLAB script provided by Piazza et al. [44], and they were controlled for non-numerical parameters, such as dot size, total occupied area, and density. The dot arrays appeared after 1000 ms and disappeared again after 840 ms to avoid counting the number of dots. A total of 36 trials were presented to the participants, each trial being initiated by the experimenter. Kindergartners were instructed to perform both accurately and quickly and to press a stickered key on the side of the larger dot array (D for the left and K for the right). Answers were registered by the computer. Task performance was mean response time divided by mean accuracy. To familiarize participants with the key assignments, three practice trials were included per task.

Symbolic Numerical Magnitude Processing

Symbolic numerical magnitude processing skills were measured with a comparison task that was the same as the nonsymbolic numerical magnitude processing, except that the dot arrays were replaced by Arabic numerals, ranging from 1 to 9. The trial sequence was the same as in the nonsymbolic comparison task, except that the stimuli remained visible until the response. Task performance was the mean response time divided by mean accuracy. Three practice trials preceded the task to familiarize participants with the key assignments.

Numeral Recognition

A numeral recognition task was used to examine whether kindergartners can already recognize Arabic numerals consisting of single digits as well as more complex numerals that consist of multiple digits [45]. Single digits were randomly presented, and kindergartners were asked to name each numeral (2—1—4; 3—7—6; 5—9—8). Subsequently, a series of complex numerals were presented, and participants were asked to name these. The complex numerals were presented in blocks of increasing difficulty (10—17—13; 11—14—18; 31—26—45; 27—56—80; 107—164—270; 1007—1052—3204; 90,080—15,029—24,356). We applied a stopping rule, and the task was terminated if a child made three consecutive errors. Each correctly recognized (single and complex) numeral was rewarded with one point, and total accuracy was used to indicate performance.

Arithmetic with Marbles and Arabic Digits

Children's early arithmetic was assessed with two arithmetic tests [46] consisting of the same addition and subtraction problems. The concrete arithmetic test was first administered: problems were presented to the children with concrete materials, namely marbles. The examiner placed several marbles in an opaque cup and told the child how many marbles were in the cup. Marbles were added or removed, and the child was told how many marbles were being added or removed.

Secondly, the symbolic arithmetic test was assessed. The symbolic test used the same problems as the concrete one but presented in Arabic numerical symbols. The problems were phrased as follows: "How much is 1 and 1?" and "How much is 2 take away 1?". Both tests consisted of two practice items ($1 + 1$, $2 - 1$) and eight problems ($3 - 1$, $2 + 4$, $6 - 4$, $4 + 3$, $2 + 1$, $5 - 2$, $3 + 2$, $7 - 3$), with sums and minuends of seven or less. Children received one point for each solved problem (maximum = 8, per arithmetic task).

Matrix Reasoning

To determine kindergartners' intellectual ability, we administered the Matrix Reasoning subtest of the Dutch Wechsler Intelligence Scale for Children, Third Edition (WISC-III-NL; Wechsler, 2011). This allowed us to investigate whether a common reliance on intellectual ability could explain the expected associations between early reading and early arithmetic.

### 2.2.2. Parents

Arithmetic

The parents' arithmetic competence was assessed with an extended version [47] of the Tempo Test Arithmetic [48]. The addition subtest, as well as the subtraction subtest of this timed achievement test, was presented. The original subtests involved 40 problems of increasing difficulty. To avoid ceiling effects in high math achievers, both these subtests were extended with 20 additional additions or subtractions. Children had to solve as many single-digit and multi-digit additions as possible within 1 min. Afterward, the children completed the subtraction subtest in the same way. The score on this test is the number of correctly solved problems on each subtest within the time limit per subtest (maximum = 120). This test combines speed and accuracy into one index score. The test manual reports that the reliability estimate of this test is 0.92.

Math Anxiety

The Mathematics Anxiety Rating Scale-Revised (MARS-R [49] was used to evaluate the parents' mathematics anxiety of the parents. This revised and shortened questionnaire from 1982 consists of 25 items which were selected from the original questionnaire, i.e., Mathematics Anxiety Rating Scale [50], consisting of 98 items. By using a five-point Likert scale, all parents were asked to indicate for each item how anxious they would be in such a situation (1 = not at all anxious, 5 = very anxious). They also had the option to respond, "not applicable." The score on this test is the average score on the Likert scale (maximum = 5), with a high score suggesting a high degree of mathematics anxiety.

Educational Level

All the parents involved in this study were asked to indicate the highest educational level that they completed. They could choose from the following response options: (1) Primary education, (2) Secondary education, (3) Higher level vocational qualification, or (4) Academic bachelor's or master's degree.

Caregiving Time

Both parents received the following question: "Who spends most time with the child?" providing an indication of who the main caregiver is. According to their own estimation, they could respond mother (=1, indicating the child spent most of their time with the mother), equal (=2, indicating the child spent the same amount of time with the mother and the father), or father (=3, indicating the child spent most of their time with the father). By calculating the sum of the responses of both parents, we created a variable that indicates the distribution of care between parents. If the mother and father in a family indicated the mother as the main caregiver, this would result in a total score of 2, versus if both parents designated the father as the main caregiver, this would result in a total score of 6. A score of 4 would indicate that the care of the child was equally distributed between both parents.

## 3. Results

### 3.1. Differences between Fathers and Mothers

The descriptive statistics for the parental data are given in Table 1.

Matched pairs *t*-tests showed that fathers scored higher than mothers at arithmetic; ($t = 2.45$; df = 82; $p = 0.016$; Cohen's $d = -0.27$). Mothers showed higher mathematics anxiety than fathers ($t = 2.78$; df = 82; $p = 0.007$; Cohen's $d = 0.305$). Though data on educational level was not available for all participants, mothers had higher educational levels than fathers ($t = -0.42$; $df = 65$; $p < 0.001$; Cohen's $d = 0.517$).

**Table 1.** Descriptive statistics on parents' educational level and scores.

|  | *n* | **Min** | **Max** | **Mean** | *SD* |
|---|---|---|---|---|---|
| **Mother** | | | | | |
| Educational level | 70 | 1 | 4 | 3.16 | 0.79 |
| Arithmetic | 83 | 0 | 86 | 63.80 | 12.16 |
| Mathematics anxiety | 83 | 25 | 118 | 58.78 | 22.09 |
| **Father** | | | | | |
| Educational level | 76 | 1 | 4 | 2.68 | 0.90 |
| Arithmetic | 83 | 0 | 88 | 67.11 | 14.34 |
| Mathematics anxiety | 83 | 25 | 129 | 49.08 | 21.45 |

## 3.2. Within-Parent Correlations

Mothers' mathematics anxiety correlated significantly with their arithmetic score ($r(83) = -0.311$; $p = 0.004$ and with their educational level ($r(70) = -0.309$; $p = 0.009$). However, their arithmetic score did not correlate significantly with their educational level ($r(70) = 0.001$; $p = 0.995$).

Fathers' mathematics anxiety correlated significantly with their arithmetic score ($r = -0.28$; $p = 0.009$) and with their educational level ($r = -0.529$; $p < 0.001$). Their arithmetic score correlated significantly with their educational level ($r = 0.324$; $p = 0.004$).

## 3.3. Spousal Correlations

Fathers' and mothers' educational levels correlated significantly ($r = 0.296$; $p = 0.016$), as did their arithmetic scores ($r = 0.58$; $p < 0.001$). Their mathematics anxiety scores did not, however, correlate significantly ($r = -0.064$; $p = 0.568$). Thus, we found assortative mating for educational level and mathematics skills but not for mathematics anxiety.

## 3.4. Children's Test Scores and Correlations

Moving on to the data in children, the descriptive statistics on children's arithmetic skills are given in Table 2. The correlations are presented in Table 3.

**Table 2.** Descriptive statistics on children's test scores.

|  | *n* | **Min** | **Max** | **Mean** | *SD* |
|---|---|---|---|---|---|
| Nonsymbolic comparison RT | 82 | 696 | 3559 | 1245 | 438 |
| Symbolic comparison RT | 82 | 670 | 6934 | 2007 | 1116 |
| Numeral recognition | 83 | 3 | 29 | 14.96 | 5.57 |
| Arithmetic-marbles | 83 | 1 | 8 | 5.73 | 1.73 |
| Arithmetic-Arabic digits | 83 | 0 | 8 | 4.27 | 2.72 |
| Arithmetic-composite score | 83 | 2 | 16 | 10 | 3.99 |
| Matrix Reasoning | 83 | 6 | 26 | 15.27 | 4.17 |

Note. RT = response time.

**Table 3.** Correlations between children's test scores.

| | | Category | | | | | |
|---|---|---|---|---|---|---|---|
| | | **Symbolic Comparison RT** | **Numeral Recognition** | **Arithmetic Marbles** | **Arithmetic Arabic Digits** | **Arithmetic Composite Score** | **Matrix Reasoning** |
| Nonsymbolic comparison RT | | | | | | | |
| | *r* | 0.17 | **−0.24** | −0.11 | −0.21 | −0.19 | −0.07 |
| | *p* | 0.14 | 0.03 | 0.33 | 0.06 | 0.09 | 0.51 |
| Symbolic comparison RT | | | | | | | |
| | *r* | | **−0.33** | −0.18 | **−0.21** | −0.20 | −0.17 |
| | ***p*** | | 0.002 | 0.10 | 0.05 | 0.08 | 0.08 |
| Numeral recognition | | | | | | | |
| | *r* | | | **0.59** | **0.62** | **0.68** | **0.31** |
| | *p* | | | <0.001 | <0.001 | <0.001 | 0.004 |
| Arithmetic marbles | | | | | | | |
| | *r* | | | | **0.59** | **0.84** | **0.35** |
| | *p* | | | | <0.001 | <0.001 | <0.001 |
| Arithmetic Arabic digits | | | | | | | |
| | *r* | | | | | **0.94** | **0.39** |
| | *p* | | | | | <0.001 | <0.001 |
| Matrix Reasoning | | | | | | | |
| | *r* | | | | | | **0.42** |
| | *p* | | | | | | <0.001 |

Note. RT = response time. Correlations that are significant ($\alpha = 0.05$) are in bold.

*3.5. Parental Characteristics and Children's Test Scores*

Mothers' educational level correlated significantly with their children's arithmetic-marbles score ($r(70) = 0.437$; $p < 0.001$); arithmetic-Arabic digits score ($r(70) = 0.329$; $p = 0.005$); and composite arithmetic score ($r(70) = 0.314$; $p = 0.008$), but not with any other child measure. Neither mothers' arithmetic score nor their mathematics anxiety correlated with any child measure, nor did any child measure correlate with their fathers' educational level, arithmetic, or mathematic anxiety score.

The mean time-spending score was 3.12 ($SD = 0.903$). Parents were highly consistent in their answers (i.e., 73% of parents rated division of care as identical to their partner). As an additional indicator of consistency in parents' response to this item, a correlation coefficient was calculated ($r = 0.71$; $p < 0.001$), indicating high consistency. In order to investigate whether relative amount of caregiving by mother and father moderated the correlation between mother's educational level and children's composite arithmetic score, the standardized caregiving scores were multiplied by the standardized maternal educational level scores to give the interaction term. Multiple regression was then carried out with composite arithmetic score as the dependent variable and mother's educational level and the interaction term as the predictor. Mother's educational level was a significant predictor ($\beta(2,67) = 0.315$; $t = 2.74$; $p = 0.008$), but the interaction term was not ($\beta(2,67) = 0.017$; $t = 0.145$; $p = 0.885$). When time-spending score (i.e., division of care), mothers' educational level, and fathers' educational level were all included as predictors in a multiple linear regression with children's composite arithmetic score as the dependent variable (Table 4), mothers' educational level continued to be a significant predictor, but neither fathers' educational level, nor time-spending score.

**Table 4.** Multiple regression analysis of children's arithmetic skills.

| Predictor | $\beta$ | $t$ | $p$ |
|---|---|---|---|
| Mother's educational level | 0.39 | 2.51 | 0.015 |
| Father's educational level | 0.11 | 0.87 | 0.390 |
| Division of care | −0.19 | −1.51 | 0.136 |

Note. The dependent variable in this regression is children's composite arithmetic score. The three predictors together explain 13.4% of the variance in children's composite arithmetic scores.

## 4. Discussion

As predicted, fathers performed somewhat better at arithmetic than mothers and also showed lower mathematics anxiety. Less predictably, mothers showed higher levels of education. This was not due to differences regarding higher-level academic versus vocational qualifications; mothers were more likely than fathers to have either. In recent years, women have been more strongly represented in higher education than men, contrasting with the situation in earlier times, and this may have affected the parents in this study.

Mathematics anxiety was associated with educational level and arithmetic in both fathers and mothers, in contrast with Vanbinst et al.'s [23] finding that it was associated with educational level in mothers but not fathers. In the present study, arithmetic was associated with educational levels in mothers but not fathers. This might reflect differences in course and career choices by men and women as regards emphasis on mathematics versus other subjects. In any case, the results, like the rather different result of Vanbinst et al.'s [23] study, the results indicate the importance of looking not only at gender differences in arithmetic or mathematics anxiety but also at gender differences in their associations with other factors.

As predicted, the mothers' and fathers' educational levels were correlated, as were their arithmetic scores, suggesting assortative mating. Our study design was not suited to investigate the reasons for this assortative mating. However, contrary to prediction and in contrast with Vanbinst et al.'s [23] findings, there was no significant correlation between fathers' and mothers' mathematics anxiety. This unexpected result may need to be taken into account when considering the relationship, or lack of it, between parental mathematics anxiety and children's mathematical performance.

The results indicated that there were significant correlations between most but not all of the children's test scores. Strikingly and contrary to predictions, reaction time for nonsymbolic comparison did not correlate significantly with the reaction time for symbolic comparison, suggesting early 'symbolic estrangement' [51]. The reaction time for nonsymbolic comparison did correlate moderately with numerical recognition and with the composite arithmetic score, and the reaction time for symbolic comparison correlated strongly with numerical recognition and with the composite arithmetic score. Matrix reasoning also correlated strongly with numerical recognition and composite arithmetic but not with reaction time for either symbolic or nonsymbolic comparison. When matrix reasoning, reaction time for symbolic comparison, and reaction time for nonsymbolic comparison were included together in multiple regressions, the latter ceased to be a significant independent predictor of either numeral recognition or composite arithmetic score, while matrix reasoning and reaction time for symbolic comparison were independent predictors of both. As expected, there were no significant gender differences in children's mathematical performance.

Contrary to predictions, neither parents' mathematics anxiety predicted their children's mathematical performance. This may indicate that such relationships do not develop until later on. However, a few studies of older children [21,23] have suggested that parental mathematics anxiety predicted children's mathematics anxiety but not their mathematical performance. Thus, one potential limitation of the present study is that no direct measure of mathematics anxiety was obtained from the children themselves. Usually, direct measures of mathematics anxiety are obtained from somewhat older children than those in the present study; it is challenging to study math anxiety in very young children. Nevertheless,

some recent studies have resulted in the development of mathematics anxiety measures for school beginners [17], and it would be desirable for future studies to include such measures.

This may not, however, fully explain the contrasting results of the few studies that have looked at associations between parental mathematics anxiety and their young children's performance. Becker et al. [40] found such an association even for preschoolers, whereas Cosso et al. [41], like the present study, did not. It is possible that different forms of mathematics anxiety, or different associations with other characteristics, may be related in different ways to children's performance. Cosso et al. [41] found two factors for parental mathematics anxiety, anxiety about their own mathematical performance and anxiety about mathematical activities with their children, though neither predicted their preschoolers' mathematical performance. Silver et al. [52] found the combination of high levels of parental mathematics anxiety and strong parental beliefs in the importance of mathematics predicted high mathematical performance by their preschool children, while the combination of high levels of parental mathematics anxiety and lack of parental beliefs in the importance of mathematics predicted low mathematical performance by their preschool children.

It may be that the influence of parents' mathematics anxiety on both their children's mathematical performance and their children's mathematics anxiety may depend on the extent to which they are involved in their children's mathematical activities. As mentioned in the introduction, Maloney et al. [22] found that parental mathematics anxiety only significantly predicted children's mathematics anxiety and performance if they also had significant involvement in their children's mathematics homework. Although the present study investigated the caring role of each parent, it did not look directly at their involvement in children's mathematics education. It may be that although parental mathematics anxiety shows an overall relationship to children's mathematics anxiety, it is only linked to performance if it is combined with heavy involvement in children's mathematical activities. More specifically, it may only be linked to performance if it is combined with heavy involvement with those mathematical activities that may arouse concerns about external evaluation, such as homework and examination preparation. Such activities may not yet be central to kindergarten children and their parents, which could be one explanation for the fact that parental mathematics anxiety does not appear to predict kindergarten children's mathematics performance. Future studies should investigate these possibilities.

The only parental characteristic that was associated with children's mathematical performance was mothers' educational level, which was correlated with children's composite arithmetic score, but not with any other child measure, including nonverbal intelligence. It could be that mothers' educational level predicts children's mathematical performance because of the home-numeracy environment they create, thus environmental transmission, or because genes that influence educational level in the mothers also influence mathematics in the children, thus genetic transmission, or a combination of both environmental and genetic transmission.

A limitation of the study is the relatively small sample size. Even though there were over 200 participants, the analytic unit is 83, so we need to be cautious in interpreting the results. Larger samples in future studies might lead to firmer conclusions and might also enable some finer-grained analyses: e.g., looking at whether the patterns of associations between parental and child characteristics might differ for boys and girls. Another limitation common to most studies of associations between parental anxiety and attitudes toward mathematics and children's mathematical performance is the difficulty of distinguishing between parental environmental influences and other factors, including genetic ones. For example, Wang et al. [53] suggested that there is about 40% heritability of mathematics anxiety. Parent–child correlations reflect a combination of genetic and environmental parent-to-child transmission. The current study, like most studies on this topic, cannot distinguish the two. Hart et al. [54] suggest some ways in which future studies might explore this issue.

Future studies should also include larger samples and, if possible, be longitudinal, to investigate whether relationships between parental mathematics anxiety and children's

mathematical performance change over time. They should also include measures of a wider variety of parents' reactions to mathematics than just anxiety, including their beliefs about the importance of mathematics [52], their gender stereotypes about mathematics, their beliefs about their own and their children's mathematical ability, and their enjoyment of mathematics. They should also include a wider variety of families, like single-parent families, adoptive families, and families with a step-parent, to investigate whether the results would differ from those for children living with both biological parents.

All such studies should be carried out with an element of caution about the nature and direction of causation between parental anxiety and attitudes toward mathematics and children's mathematical performance. Any such relationships do not necessarily mean that parental attitudes influence children's performance. They could reflect genetic influences (e.g., [53] suggested that there is about 40% heritability of mathematics anxiety) or some environmental factor affecting mathematics anxiety and arithmetical performance in both parents and children.

**Author Contributions:** Conceptualization, E.B. and E.v.B.; methodology, E.B.; formal analysis, E.B. and A.D.D.; investigation, E.B.; data curation, E.B., E.v.B. and A.D.D.; writing—original draft preparation, A.D.D.; writing—review and editing, E.B., E.v.B. and A.D.D.; project administration, E.B. All authors have read and agreed to the published version of the manuscript.

**Funding:** The data collection was funded by a postdoctoral fellowship (grant number 12N0617N) of the Research Foundation Flanders (FWO) to Kiran Vanbinst. Elien Bellon is a postdoctoral fellow supported by KU Leuven (grant number PDM/20/057) and FWO (grant number 12C9523N). Elsje van Bergen is a Jacobs Foundation Fellow and is further supported by ZonMw grant 531003014 and NWO Gravitation grant 024.001.003.

**Institutional Review Board Statement:** The study was conducted in accordance with the Declaration of Helsinki. The study and consent procedures were approved by the Social and Societal Ethics Committee of the University of Leuven, Belgium (G2016 03 533).

**Informed Consent Statement:** Informed consent was obtained from all subjects involved in the study.

**Data Availability Statement:** The data presented in this study are available on request from the corresponding author. The data are not publicly available due to privacy reasons.

**Acknowledgments:** We thank Kiran Vanbinst for conceptualizing and running the data collection. We would also like to thank all participating families.

**Conflicts of Interest:** The authors declare no conflict of interest.

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
