# Peer review of "Is Parental Mathematics Anxiety Associated with Young Children’s Arithmetical Performance?"

_education, doi:10.3390/educsci12110812_

Round 1
Reviewer 1 Report
Please find my comments in the file attached.

Reviewer 2 Report
Overall this article is a strong fit for educational sciences. The article is detailed, clear, and builds upon existing literature to push our understanding of math anxiety forward. As the authors acknowledge, one of the biggest limitations of this article is the small sample size of 83 mother-father-child trios. While the author's acknowledge this, sharing how many other participants they collected I believe would make this part of the paper stronger. For example, was data also collected from single parents, adoptive families, LGBTQ families, and grandparent families?
Additionally, the article mentions math avoidance as one possible mechanism by which mathematics anxiety and mathematical performance interact, but does not provide a citation for this idea. The article by Choe et al., (2019) might further support their argument (not my publication). I think this may help explain why mothers’ educational levels continued to be a significant predictor on children's test scores.
Further explaining why the authors expected parents' math anxiety to correlate with some of the numerical skills listed would strengthen the methods section, since these seem like very distinct constructs that may be more or less impacted by preschool programs.
Finally, there is likely a significant difference between the math input parents share with preschoolers/ kindergarteners compared to with later elementary students. l I believe this is one possibility for the variety of findings about the relation between parents' (typically mothers') math anxiety and children's math performance. I believe the Maloney et al., (2015) citations provided highlight this finding, with the focus on homework.
